# COVID-19 Anti-Vaccine Sentiments: Analyses of Comments from Social Media

**DOI:** 10.3390/healthcare9111530

**Published:** 2021-11-09

**Authors:** Li Ping Wong, Yulan Lin, Haridah Alias, Sazaly Abu Bakar, Qinjian Zhao, Zhijian Hu

**Affiliations:** 1Centre for Epidemiology and Evidence-Based Practice, Department of Social and Preventive Medicine, Faculty of Medicine, University of Malaya, Kuala Lumpur 50603, Malaysia; haridahalias@gmail.com; 2Department of Epidemiology and Health Statistics, School of Public Health, Fujian Medical University, Fuzhou 350122, China; yulanlin@fjmu.edu.cn; 3Tropical Infectious Diseases Research and Educational Centre (TIDREC), University of Malaya, Kuala Lumpur 50603, Malaysia; sazaly@um.edu.my; 4Department of Medical Microbiology, Faculty of Medicine, University of Malaya, Kuala Lumpur 50603, Malaysia; 5State Key Laboratory of Molecular Vaccinology and Molecular Diagnostics, National Institute of Diagnostics and Vaccine Development in Infectious Diseases, School of Public Health, Xiamen University, Xiamen 361005, China; qinjian_zhao@xmu.edu.cn

**Keywords:** antivaccine, social media, vaccine hesitancy

## Abstract

Purpose: This study analyzed the insights and sentiments of COVID-19 anti-vaccine comments from Instagram feeds and Facebook postings. The sentiments related to the acceptance and effectiveness of the vaccines that were on the verge of being made available to the public. Patients and methods: The qualitative software QSR-NVivo 10 was used to manage, code, and analyse the data. Results: The analyses uncovered several major issues concerning COVID-19 vaccine hesitancy. The production of the COVID-19 vaccine at an unprecedented speed evoked the fear of skipping steps that would compromise vaccine safety. The unknown long-term effects and duration of protection erode confidence in taking the vaccines. There were also persistent concerns with regard to vaccine compositions that could be harmful or contain aborted foetal cells. The rate of COVID-19 death was viewed as low. Many interpreted the 95% effectiveness of the COVID-19 vaccine as insufficient. Preference for immunity gains from having an infection was viewed as more effective. Peer-reviewed publication-based data were favoured as a source of trust in vaccination decision-making. Conclusions: The anti-COVID-19 vaccine sentiments found in this study provide important insights for the formulation of public health messages to instill confidence in the vaccines.

## 1. Introduction

The SARS-CoV-2 vaccine development efforts began with the onset of the novel coronavirus outbreak, which first emerged in Wuhan, China, in late December 2019. The coronavirus disease 2019 (COVID-19) eventually became an ongoing pandemic with no specific treatments or vaccines available for prevention. The COVID-19 vaccine is likely to be the most effective and sustainable approach for controlling the pandemic. Unprecedented research effort and global coordination resulted in the rapid development of vaccine candidates and the initiation of many clinical trials worldwide. As of 12 November 2020, according to the World Health Organisation’s draft landscape of COVID-19 vaccines, there were 48 candidate COVID-19 vaccines in clinical trial evaluation and 164 in preclinical evaluation stages [1]. The acceptance of a COVID-19 vaccine has been reported in several studies worldwide. An early study of COVID-19 vaccine acceptance in France conducted 10 days after the nationwide lockdown revealed 74% acceptance [2]. A higher acceptance rate was found among the Asian population. Evidence from two studies from Southeast Asian countries conducted in April 2020 reported higher COVID-19 vaccine acceptance rates of 94.3% [3] and 93.3% [4], respectively. A large sample study in China reported 83.3% acceptance among the public in China in May 2020 [5]. A relatively lower acceptance rate (67.0%) was reported in a study in the U.S. conducted in May 2020 [6]. A recently published global survey of 19 countries conducted in June 2020 reported acceptance rates ranging from 54.95% to 88.6%, with the lowest reported in Russia and the highest in China [7]. Potential adverse events and negative consequences from COVID-19 trials have begun to emerge in various news media outlets since May 2020, refs. [8,9,10,11,12] although published clinical evidence from COVID-19 trials has indicated that the vaccines are well-tolerated with mild or moderate severity adverse effects [13,14]. A recently conducted survey in October 2020 in the US showed a further decline in people’s willingness to take the COVID-19 vaccine, with only 51% having expressed vaccination intention [15].

Vaccine hesitancy is a growing threat to global health security. The World Health Organization has named vaccine hesitancy as one of the top 10 threats to global health in 2019 [16]. Tracking public responses to COVID-19 vaccination is crucial to understanding any concerns and acceptance regarding the forthcoming vaccines. Often, anti-vaccine viewpoints are widespread in social media [17]. Of late, there has been remarkable popularity in international news channels broadcasting using social media platforms such as Facebook, Twitter, and Instagram. As global social media usage continues to grow, the potential to harness data generated from social media users has also grown [18]. In particular, the activities of social media users, by commenting or messaging, serve as new research avenues with which to harvest empirical insights on public perception. We, therefore, examined public opinion on social media following the first press releases of data showing the high effectiveness of COVID-19 vaccines and the announcement that they would soon be available. 

## 2. Material and Methods

### 2.1. Data Collection

We extracted users’ comments on international television channels’ (namely, BBC World News and CNN International) postings on Instagram and Facebook on 16 November 2020: (1) US COVID-19 vaccine nearly 95% effective, early data shows, (2) COVID-19 vaccine: will young people take it? (3) Moderna’s coronavirus vaccine is 94.5% effective, according to company data and (4) Moderna: COVID vaccine shows nearly 95% protection. The comments from these posting were selected as they were the first public release news announcing the effectiveness of the vaccines and that they would soon be made available to the public. All comments were followed up daily. The number of comments ceased rapidly a day after posting and on the third day, no new themes were being identified. All cumulative postings were copied on 18 November 2020. In total, 3652 comments from the two BBC postings and 1728 comments from CNN were extracted from Instagram. A total of 4325 comments from a BBC posting on Facebook were extracted. In the analyses, comments about the politicisation of the COVID-19 vaccine were not included. Comments in languages other than English were also not included in the analyses. All of the comments extracted for analysis in this study were publicly available through Instagram and Facebook, so informed consent was not required.

### 2.2. Data Analysis

NVivo 12 software (QSR International Pty Ltd., Doncaster, Australia) was used to conduct a thematic analysis of the comments. Textual content from the comments that contained anti-vaccination sentiments were coded. The thematic analysis included reading each comment closely, identifying patterns, assigning codes, and formulating themes and sub-themes from the data [19]. To ensure that optimal analytical rigour was practised, the data were analysed and coded independently by the researchers (WLP and HA), after which they were scrutinised, compared, and discussed. Any discrepancies were resolved through a consensus discussion with a third person.

## 3. Results

Analysis of the comments yielded eight central themes surrounding COVID-19 anti-vaccine sentiments: (1) the rush for a COVID-19 vaccine, (2) unknown long-term effects of a new vaccine, (3) unknown duration protection of a new vaccine, (4) transparency of vaccine compositions, (5) new mRNA technology, (6) preferences for natural immunity, (7) vaccine effectiveness versus survival rate, and (8) a lack of scientific evidence. Figure 1 illustrates the themes and sub-themes identified in the data. 

### 3.1. Rush for COVID-19 Vaccine

Many comments surrounded the issues of the COVID-19 vaccines being rushed through the clinical phase. The accelerated speed of the development has caused concern that vaccines might be approved with many steps skipped. Indicative is the following quote:


*“I cannot trust a vaccine produced this quick. So many steps skipped.”*



*“I cannot trust a vaccine produced this quick. So many steps skipped and we don’t know if long-term side effects are a thing.”*


Many believed that no vaccine in history has been developed at such an unprecedented speed and feared that critical steps were skipped during the development process that safety and efficacy were overlooked. Many also noted that despite having full faith in the efficacy of vaccines, being pro-vaccine, or even noting that they had never missed any recommended vaccines, the fact that the COVID-19 vaccine was being released so quickly induced the greatest reluctance for them to accept the vaccine. There was also a concern that the COVID-19 vaccine has not been tested on an adequate number of subjects in clinical trials. Other anecdotes for the rush were numerous, including being politically and profit-motivated.


*“It’s come out so fast that it’s clearly politically and profit-motivated.”*



*“Has this vaccine undertaken rigorous testing with a large sample group?”*



*“The real question is how many people did they test this vaccine on.”*


### 3.2. Long-Term Effect

Worry about the long-term effects of new vaccines were paramount. Many believe that the vaccine has not yet been perfected and that only time will tell if it has serious side-effects.


*“It is not being an anti-vaxxer; it’s called making a sensible educated decision. Vaccines usually take 7 years to develop. I’m not willing to be a guinea pig for pharmaceutical companies”*



*“How can anyone trust a vaccine without any long-term trials?”*



*“Vaccines usually take a decade or more of research. No one knows what the side effects may be of an expedited vaccine. I’m pro-vaccination but not for one that was pushed through research.”*



*“As soon as it’s proven safe with longer-term trials, I’d be perfectly happy to take it, but with the long-term side effects being so unknown, I’d be very nervous to.”*



*“It takes years for side effects to appear, and they could very well be worse than the sickness.”*


There was heightened concern that the COVID-19 vaccine would cause death or infection. Common anti-vaccination beliefs, such as vaccination cause autism and chronic conditions, were also raised. A young woman expressed concern regarding the safety of the vaccine with a particular focus on whether it could cause infertility. 


*“I’m trying to get pregnant, has it been tested to ensure that it will not affect my reproductive organs or compromise future pregnancy. The vaccine is just too new. We don’t know the long-term side effects.”*


Many comments indicated that they would wait for a certain period after the vaccines have been released onto the market. There were mixed comments regarding how long they would wait, with the duration of waiting ranging from 6 months to 5 years. 


*“I would wait 6 months to see how it affects people.”*



*“If I was to get vaccinated it’s going to be in five years. Many things could go wrong such as late side effects. When people get a high dose of radiation, they don’t get cancer straight away it takes time.”*



*“There are no credible data to prove the long-term safety and efficacy of this vaccine. At this stage, it’s still experimental, even if all regulatory approvals are in place.”*


### 3.3. Duration of Protection

Many desire information about the duration of protection of the new COVID-19 vaccine and whether a booster dose is needed following primary vaccination, similar to the influenza vaccines. The fact that the influenza virus vaccination is recommended annually meant that many expressed concern around whether revaccinations would also be needed for the COVID-19 vaccines. Furthermore, there was also concerns that if the SARS-CoV-2 virus is rapidly-mutating, similar to the influenza viruses, the current COVID-19 vaccine which is under development will no longer be effective when it becomes available.


*“My only concern really... how long is it in your body? Is it like flu vaccine and you need a new one each year?*



*“But how long is the vaccine effective?”*



*“Does the constant mutation impact its efficacy?”*



*“The virus is a mutating one, as previously said; how will a vaccine be of any use anyhow? The virus will mutate, become stronger & still infect people.”*



*“This is not possible because the virus has mutated into different strains.”*


### 3.4. Vaccine Compositions 

Issues concerning the vaccine compositions were raised. Many questioned the full ingredients of the COVID-19 vaccines. 


*“What is in the vaccine?”*



*“We need to know what the components are that make up this particular vaccine given the stigma and hesitation from so many citizens of the World.”*


Worries are mounting that the COVID-19 vaccine contains the mercury-based chemical thimerosal, similar to many other vaccines, and that the vaccination can cause autism. Vaccine sceptics have posted the comment: “*Research the thousands of hidden VACCINE INJURIES that have taken the lives of many or left them with lifelong damage*”, and subsequently provided a long list of links to scientific publications on the association between the mercury containing preservative thimerosal in vaccines and autism, warning others to check the ingredients in COVID-19 vaccines. There were comments that the COVID-19 vaccine contains “aborted human foetuses”. Some expressed a moral disgust over the use of aborted foetal cells in the development of COVID-19 vaccines. Those who have egg allergies particularly want to know whether the COVID-19 vaccine contains egg proteins and emphasise that the full components should be made known for them to decide whether or not to be vaccinated.


*“Is it likely to contain Mercury and other heavy metals? What’s the risk as with other vaccines of other complications”*



*“There’s aborted babies in this new vaccine... ”*



*“Anything that would have aborted foetal tissue in it I’m not putting in my body.”*



*“The vaccine probably contains mercury and other chemicals including raw eggs.”*


### 3.5. New mRNA Vaccine Technology

Various worries concerning the use of messenger RNA (mRNA) in vaccine production were raised. Many worried about the unknown risks to mRNA vaccines that is based on new technology. Some associated the use of nanotechology in vaccine development with “microchip” conspiracy. They claimed that the COVID-19 vaccination is an attempt to implant humanity with microchips under the guise of vaccination. Many noted that they do not want their body to be injected with “nano chips” or “DNA microchips” that will alter or modify human DNA.


*“An mRNA vaccine a year ago was only “theoretical”; definitely hasn’t been around long enough to have any long-term studies.”*



*“What about the conspiracy theory that it’s a way of injecting nano chips...?”*



*“This vaccine is part of their agenda. Nobody wants a microchip.”*



*“The new vaccine for COVID-19 will be the first of its kind EVER. It will be an mRNA vaccine which will literally alter your DNA. “*


### 3.6. Preference for Natural Immunity

There was also a belief that immunity from naturally acquired SARS-CoV-2 infection is better than that from a vaccine. Naturally acquired immunity was viewed as highly effective in protection against subsequent reinfections. In contrast, immunity acquired by vaccination was regarded as risky.


*“Nature immunity, artificial immunity comes with risk.”*



*“Natural immunity is 99.9% effective.”*



*“My immune system is 99.4% effective…”*


### 3.7. Vaccine Effectiveness versus Survival Rate

Some rejected the COVID-19 vaccines because COVID-19 has a “high survival rate” and outweighs the risk of taking a new vaccine. Many tend to compare the relatively higher survival rate of over 99% in contrast to the only 95% COVID-19 vaccine effectiveness. The high survival rate implies that vaccination is unnecessary, considering many uncertainties in the new COVID-19 vaccines.


*“It’s not worth risking taking it. It’s worse than coronavirus itself.”*



*“They (young people) don’t need it as it doesn’t affect the young.”*



*“99.6% survival rate sounds like the opposite of deadly in my opinion.”*



*“You need a vaccine for something that you have a 1% chance of getting? And a 99% chance of full recovery?”*



*“A vaccine for a virus that has a death rate of 1%? I’ll take my chances.”*



*“95% effective for a virus that kills at 0.5%.”*


There was also an opinion that 95% effectiveness of the COVID-19 vaccine is insufficient, making it unsafe to be administered, and warrants a COVID-19 vaccine with a greater effectiveness figure. 


*“If it’s 95% effective, what are the consequences to the remaining 5%? Is it side effects or death? Honestly, it has to be 100%. 95% is not yet perfect because in 100 people 5 will still be get (the infection)..?”*



*“How long can this take to be verified as safe? 94.6% isn’t safe to me.”*



*“A 94.5% effective, brand new vaccine with unknown long-term effects versus risking getting a virus that has a 99.96% survival rate and then building up natural immunity anyway. I’ll take my chances.”*


### 3.8. Lack of Scientific Evidence

Many have little confidence in the reported level of effectiveness which was announced through the press, ahead of being peer-reviewed or published in medical journals. Many noted that the results of the COVID-19 vaccine clinical trials are lacking in scientific merit and warrant evidence-based findings for conclusion.


*“According to company data...We still don’t know enough about potential side effects, but I do understand the urgency.”*



*“Don’t believe it until peer-reviewed”*



*“No scientist has peer-reviewed the study, and there were no people with any illnesses included or any older people; just the healthiest people were in this study.”*



*“According to company data” which means that it’s not a reliable source.”*



*“When the stakes are so high and the profit is exorbitant, human life is of no value!”*


## 4. Discussion

Despite the enormous global effort to develop a vaccine for COVID-19 as rapidly as possible, the COVID-19 vaccine is not spared from skepticism. This study uncovered a broad range of barriers to COVID-19 vaccine acceptance that may hinder the aim of achieving herd immunity through vaccination.

The urgency of having a vaccine in response to the unprecedented COVID-19 pandemic has resulted in many academic institutions and pharmaceutical industries expediting their respective vaccine development. Nevertheless, the rush for a vaccine created fear among some members of the public as many believed that speeding up the process may imply skipping essential steps, which undermine public confidence in the vaccines. This suggests that it is important to improve communication to debunk the fear among the public surrounding the expedited COVID-19 vaccine development timelines. Furthermore, the public needs to be convinced that the production of COVID-19 vaccines in such a short time does not imply skipping steps that would compromise safety. Researchers are accelerating the entire development process due to the catastrophic impact of the pandemic. The new mRNA vaccine technology enables rapid development and large-scale production of the vaccines in an expedited timeframe alternative to conventional vaccine approaches [20]. Nonetheless, it remains a great challenge in introducing the new mRNA technology that neither has been used in commercially available vaccines nor tested in large-scale human trials [20,21]. The benefits of the mRNA vaccine along with its safety [20,22,23,24] should be highlighted to demystify the unfounded conspiracy theories, especially with regard to misinformation about mRNA vaccines potentially altering or resulting in human genetic modification.

It remains a challenge to convince the public to accept a new vaccine with the unknown long-term sequelae of vaccination and duration of protection. As most of the concern regarding the long-term side effects of the vaccine surrounds the issues of the constituent of the vaccines, it is of utmost importance to ensure the public about the safety record of the vaccine constituents. The vaccine compositions should be made known to gain the public’s trust in its safety and to address the concern of people who are allergic to certain vaccine constituents, for example, egg proteins. Likewise found in this study, the use of thimerosal, a mercury-based preservative, has been subjected to intense concern as it was thought to cause autism. Mounting evidence shows that the amount of thimerosal in vaccine is small and the risks of serious complications from preventable infections outweigh the risks of its adverse consequences [25]. Hence, the introduction of the COVID-19 vaccines should also be accompanied by messages to demystify the rumors around the mercury-based vaccine preservative. Additionally, hesitancy is also rooted in the ethical concerns of the use of human fetal cells in COVID-19 vaccine development. People with religious convictions expressed hesitancy about COVID-19 vaccines due to ethical concerns that human fetal cells were used in its development [26]. This finding highlight the key role of faith leaders as they are highly trusted individuals in communities [26].

Most importantly, the safety of the vaccine composition needs to be highlighted during vaccine promotion [27]. Another stream of concern identified is the belief that protection after vaccination may be inferior to that acquired after a natural infection. Uncertainties surrounding vaccine protection is one of the fundamental new vaccine development conundrums yet to be adequately addressed. It is currently unknown if any of the candidate COVID-19 vaccines will elicit a better immune protection response than that of the natural infection. Relatively little has been published comparing vaccination-acquired immunity versus immunity acquired following natural infection. To date, there is insufficient evidence comparing the immunity acquired following vaccination versus natural infection. While preliminary results from the ongoing vaccine clinical trials suggest good protection against SARS-CoV-2 infection, it is not yet established whether vaccination may also prevent transmission of the virus. Similarly, it is not known if the acquisition of a natural infection would limit transmission. Nevertheless, the public should be enlightened that vaccines in general have been largely successful in eradicating many infectious disease threats and that the risks of building up immunity to the coronavirus through natural infection outweigh the risks of immunisation. While studies on anti-SARS-CoV-2 antibodies are still ongoing, current evidence is suggestive that protective immunity against SARS-CoV-2 infection is short-lasting [28,29], hence, there is little evidence supporting the benefit of reliance solely on natural immunity in protection against COVID-19. 

The mutations of SARS-CoV-2 have been speculated to adversely affect the efficacy of most vaccines, resulting in fear around the need for repeat vaccination, similar to that of seasonal influenza vaccines. Nonetheless, recent evidence indicates that vaccines are unlikely to be affected by the ‘D614G’ mutation (Aspartate-to-Glycine change at position 614) of the SARS-CoV-2 spike protein [30]. Engagement with vaccine sceptics highlighting the well-established scientific data is important as they may lack access to trusted sources of vaccine information, hence, becoming more susceptible to anti-vaccine theories [31]. Highlighting these evidence-based facts will be useful to counteract the misinformation.

According to a recent meta-analysis, the global aggregated estimate of COVID-19 infection fatality rate was 0.68% (0.53–0.82%) [32]. Furthermore, claims circulating on social media stating the CDC reported that the coronavirus has a high survival rate of 99.997%, 99.98%, 99.5%, and 94.6% for the age groups of 0–19, 20–49, 50–60, and 70 years old and above; this has perhaps undermined the public perception of the severity of COVID-19. This is especially misleading if the mortality is calculated simply over the total number of cases ignoring the high mortality rates among those that are vulnerable, such as the elderly and those with co-morbidities. The public should be enlightened that measuring the impact of COVID-19 goes beyond mortality statistics [33]. The severe effects of SARS-CoV-2 infection should not be underestimated although the survival rate is high among those who are otherwise healthy and receive sufficient medical care. On the other hand, the long-term health consequences of SARS-CoV-2 infection remain unknown. Significant pulmonary sequelae such as persistent respiratory symptoms and lung abnormalities have been evidenced months after infection with SARS-CoV-2 [34,35,36] Evidence of post-recovery negative effects in younger patients implies that the potential long-term effects of COVID-19 should be a concern, even for children [37].

The public should also be made aware that no vaccine confers 100% protection against SARS-CoV-2. A recent study reported that the COVID-19 vaccine has to have an efficacy of at least 70% to 80% to end the pandemic [38]. Lastly, as trustworthy information is warranted, providing research-based evidence of vaccine effectiveness would greatly improve COVID-19 vaccine confidence and reduce controversies. Findings of the COVID-19 vaccine clinical trials should be published in order to cease the widespread concerns about the safety and effectiveness of the vaccine. Concerted efforts of the pharmaceutical industries and health authorities on local, national, and international levels are needed for improving or restoring vaccine confidence [39].

Our study has important limitations. We only recorded comments from Instagram and Facebook users, so these may not be representative of viewpoints from social media in general. Furthermore, only comments in English were analysed and therefore may not be representative of global responses. It is also important to note that the anti-vaccine sentiments found in this study were from virtual communities and do not represent those of communication in the non-social media; therefore, they may not be representative of the general population’s viewpoints. Despite these limitations, the perspectives gathered from this study were from diverse social media users. Lastly, the study has the limitation of being a qualitative study where generalisation is limited.

## 5. Conclusions

Intense anti-vaccine sentiments were heard as soon as the news about the COVID-19 vaccine’s imminent availability for use and effectiveness were made public. Such sentiments potentially have negative impacts and erode confidence in COVID-19 vaccines, which may ultimately undermine efforts to fight the pandemic. Mistrust towards COVID-19 vaccines represents a significant challenge in achieving the vaccination coverage needed to achieve population immunity. This study identified important issues surrounding COVID-19 vaccine controversies and mistrusts that may hamper its acceptance. COVID-19 vaccination interventions should be equipped to handle the identified concerns and to overcome anti-vaccination tendencies. The findings provide important implications for public health communications to combat COVID-19 vaccine hesitancy. 

## Figures and Tables

**Figure 1 healthcare-09-01530-f001:**
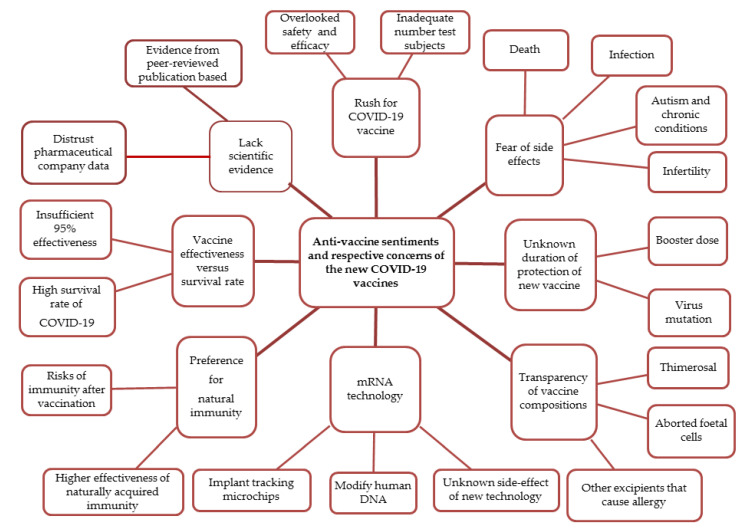
Anti-vaccine sentiments and respective concerns of the new COVID-19 vaccines.

## Data Availability

No new data were created or analyzed in this study. Data sharing is not applicable to this article.

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
