# Peer review of "COVID-19 Anti-Vaccine Sentiments: Analyses of Comments from Social Media"

_healthcare, 2021, doi:10.3390/healthcare9111530_

Round 1
Reviewer 1 Report
The article is a qualitative research paper. As the authors say, according to the WHO, the anti-vaccine movement is one of the main threats to global public health. For this reason, it is necessary to know the arguments used and the issues that concern the population.
- The software used and the people who performed the data extraction have been explained in the methodology. Was there any protocol for data extraction? If yes, please describe it briefly.
- One issue that arises is the question of the use of abortion, which has been included in the excipient. It can be considered misclassified. It would be more of a religious aspect. The authors should not put the use of thiomersal and cell lines from abortions for vaccine development in the same category. It should be taken out of the Transparency and excipients category and included in one that is moral or religious concerns. In any case, the authors should take these aspects into account in the discussion. The Catholic Church, generally reluctant to get involved in scientific debate. This covid-19 vaccine and abortion issue has had such a significant impact on public opinion that it has forced the Vatican to take a position and publish a document. https://www.vatican.va/roman_curia/congregations/cfaith/documents/rc_con_cfaith_doc_20201221_nota-vaccini-anticovid_en.html
- . The authors state that they performed the statistical analysis. Still, there are not any statistical analysis “Author Contributions: (…)LPW and HA performed the statistical analysis.” Maybe this is a typo that the authors should correct.
Reviewer 2 Report
I have read this paper with interest, although this rather confirms anticipated findings.
The abstract (and neither the title) however does not sufficiently well reflect the overall limitations of the analyses: restricted to English only (although the introduction provides evidence that this is a global issue), related to specific BBC posting and CNN comments following the first public press releases (? Likely of the companies involved), extracted from Instagram and Facebook respectively.
A second line of comments relates to the qualification of ‘anti-vaccination sentiments’ ? as (cfr results section) issues related to theme 2, theme 3 and theme 5 eg were at launch (and not based on authority assessment, but based on marketing public press releases) are reasonable and have evolved in the meanwhile (eg repeat vaccinations have been added to the leaflet). I think that ‘concerns (cf figure 1)’ are not only a more neutral, but likely also more accurate wording on the themes as selected.
Round 2
Reviewer 2 Report
no additional comments